# Extracting low-dimensional dynamics from multiple large-scale neural population recordings by learning to predict correlations

**Marcel Nonnenmacher[1], Srinivas C. Turaga[2]** and **Jakob H. Macke[1]**[*]
[1]research center caesar, an associate of the Max Planck Society, Bonn, Germany
[2]HHMI Janelia Research Campus, Ashburn, VA
`marcel.nonnenmacher@caesar.de, turagas@janelia.hhmi.org`
`jakob.macke@caesar.de`

## Abstract

A powerful approach for understanding neural population dynamics is to extract low-dimensional trajectories from population recordings using dimensionality reduction methods. Current approaches for dimensionality reduction on neural data are limited to single population recordings, and can not identify dynamics embedded across multiple measurements. We propose an approach for extracting low-dimensional dynamics from multiple, sequential recordings. Our algorithm scales to data comprising millions of observed dimensions, making it possible to access dynamics distributed across large populations or multiple brain areas. Building on subspace-identification approaches for dynamical systems, we perform parameter estimation by minimizing a moment-matching objective using a scalable stochastic gradient descent algorithm: The model is optimized to predict temporal covariations across neurons and across time. We show how this approach naturally handles missing data and multiple partial recordings, and can identify dynamics and predict correlations even in the presence of severe subsampling and small overlap between recordings. We demonstrate the effectiveness of the approach both on simulated data and a whole-brain larval zebrafish imaging dataset.

## 1 Introduction

Dimensionality reduction methods based on state-space models [1, 2, 3, 4, 5] are useful for uncovering low-dimensional dynamics hidden in high-dimensional data. These models exploit structured correlations in neural activity, both across neurons and over time [6]. This approach has been used to identify neural activity trajectories that are informative about stimuli and behaviour and yield insights into neural computations [7, 8, 9, 10, 11, 12, 13]. However, these methods are designed for analyzing one population measurement at a time and are typically applied to population recordings of a few dozens of neurons, yielding a statistical description of the dynamics of a small sample of neurons within a brain area. How can we, from sparse recordings, gain insights into dynamics distributed across entire circuits or multiple brain areas? One promising approach to scaling up the empirical study of neural dynamics is to *sequentially* record from multiple neural populations, for instance by moving the field-of-view of a microscope [14]. Similarly, chronic multi-electrode recordings make it possible to record neural activity within a brain area over multiple days, but with neurons dropping in and out of the measurement over time [15]. While different neurons will be recorded in different sessions, we expect the underlying dynamics to be preserved across measurements.

The goal of this paper is to provide methods for extracting low-dimensional dynamics shared across multiple, potentially overlapping recordings of neural population activity. Inferring dynamics from

---

[*]current primary affiliation: Centre for Cognitive Science, Technical University Darmstadt

such data can be interpreted as a missing-data problem in which data is missing in a structured manner (referred to as 'serial subset observations' [16], SSOs). Our methods allow us to capture the relevant subspace and predict instantaneous and time-lagged correlations between all neurons, even when substantial blocks of data are missing. Our methods are highly scalable, and applicable to data sets with millions of observed units. On both simulated and empirical data, we show that our methods extract low-dimensional dynamics and accurately predict temporal and cross-neuronal correlations.

**Statistical approach:**   The standard approach for dimensionality reduction of neural dynamics is based on search for a maximum of the log-likelihood via expectation-maximization (EM) [17, 18]. EM can be extended to missing data in a straightforward fashion, and SSOs allow for efficient implementations, as we will show below. However, we will also show that subsampled data can lead to slow convergence and high sensitivity to initial conditions. An alternative approach is given by subspace identification (SSID) [19, 20]. SSID algorithms are based on matching the moments of the model with those of the empirical data: The idea is to calculate the time-lagged covariances of the model as a function of the parameters. Then, spectral methods (e.g. singular value decompositions) are used to reconstruct parameters from empirically measured covariances. However, these methods scale poorly to high-dimensional datasets where it impossible to even construct the time-lagged covariance matrix. Our approach is also based on moment-matching – rather than using spectral approaches, however, we use numerical optimization to directly minimize the squared error between empirical and reconstructed time-lagged covariances without ever explicitly constructing the full covariance matrix, yielding a subspace that captures both spatial and temporal correlations in activity.

This approach readily generalizes to settings in which many data points are missing, as the corresponding entries of the covariance can simply be dropped from the cost function. In addition, it can also generalize to models in which the latent dynamics are nonlinear. Stochastic gradient methods make it possible to scale our approach to high-dimensional ($p = 10^7$) and long ($T = 10^5$) recordings. We will show that use of temporal information (through time-lagged covariances) allows this approach to work in scenarios (low overlap between recordings) in which alternative approaches based on instantaneous correlations are not applicable [2, 21].

**Related work:**   Several studies have addressed estimation of linear dynamical systems from subsampled data: Turaga et al. [22] used EM to learn high-dimensional linear dynamical models form multiple observations, an approach which they called 'stitching'. However, their model assumed high-dimensional dynamics, and is therefore limited to small population sizes ($N \approx 100$). Bishop & Yu [23] studied the conditions under which a covariance-matrix can be reconstructed from multiple partial measurements. However, their method and analysis were restricted to modelling time-instantaneous covariances, and did not include *temporal* activity correlations. In addition, their approach is not based on learning parameters jointly, but estimates the covariance in each observation-subset separately, and then aligns these estimates *post-hoc*. Thus, while this approach can be very effective and is important for theoretical analysis, it can perform sub-optimally when data is noisy. In the context of SSID methods, Markovsky [24, 25] derived conditions for the reconstruction of missing data from deterministic univariate linear time-invariant signals, and Liu et al. [26] use a nuclear norm-regularized SSID to reconstruct partially missing data vectors. Balzano et al. [21, 27] presented a scalable dimensionality reduction approach (GROUSE) for data with missing entries. This approach does not aim to capture temporal corrrelations, and is designed for data which is missing at random. Soudry et al. [28] considered population subsampling from the perspective of inferring functional connectivity, but focused on observation schemes in which there are at least some simultaneous observations for each pair of variables.

## 2   Methods

### 2.1   Low-dimensional state-space models with linear observations

**Model class:**   Our goal is to identify low-dimensional dynamics from multiple, partially overlapping recordings of a high-dimensional neural population, and to use them to predict neural correlations. We denote neural activity by $\mathcal{Y} = \{\mathbf{y}_t\}_{t=1}^T$, a length-$T$ discrete-time sequence of $p$-dimensional vectors. We assume that the underlying $n$-dimensional dynamics $\mathbf{x}$ linearly modulate $\mathbf{y}$,

$$\mathbf{y}_t = C\mathbf{x}_t + \varepsilon_t, \qquad\qquad \varepsilon_t \sim \mathcal{N}(0, R) \qquad\qquad (1)$$
$$\mathbf{x}_{t+1} = f(\mathbf{x}_t, \eta_t), \qquad\qquad \eta_t \sim p(\eta), \qquad\qquad (2)$$

with diagonal observation noise covariance matrix $R \in \mathbb{R}^{p \times p}$. Thus, each observed variable $\mathbf{y}_t^{(i)}$, $i = 1, \ldots, p$ is a noisy linear combination of the shared time-evolving latent modes $\mathbf{x}_t$.

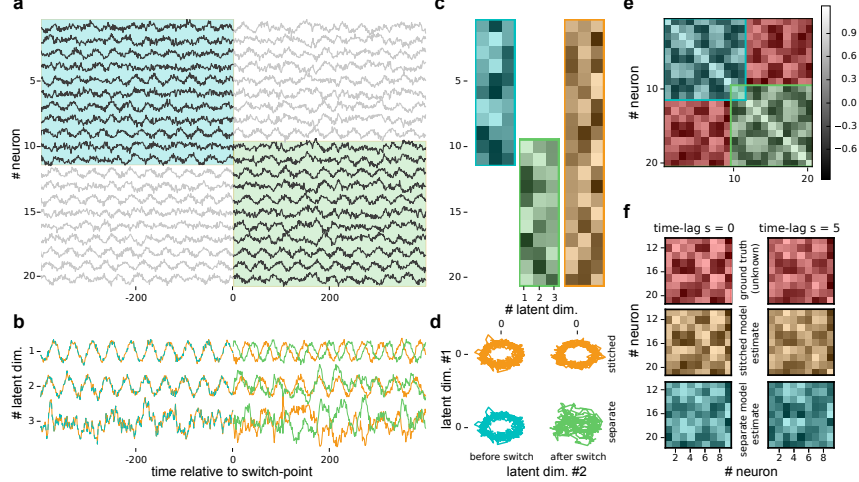

Figure 1: **Identifying low-dimensional dynamics shared across neural recordings a)** Different subsets of a large neural population are recorded sequentially (here: neurons 1 to 11, cyan, are recored first, then neurons 10 to 20, green). **b)** Low-dimensional ($n = 3$) trajectories extracted from data in **a**: Our approach (orange) can extract the dynamics underlying the entire population, whereas an estimation on each of the two observed subsets separately will not be able to align dynamics across subsets. **c)** Subspace-maps (linear projection matrices $C$) inferred from each of the two observed subsets separately (and hence not aligned), and for the entire recording. **d)** Same information as in **b**, but as phase plots. **e)** Pairwise covariances– in this observation scheme, many covariances (red) are unobserved, but can be reconstructed using our approach. **f)** Recovery of unobserved pairwise covariances (red). Our approach is able to recover the unobserved covariance across subsets.

We consider stable latent zero-mean dynamics on $\mathbf{x}$ with time-lagged covariances $\Pi_s := \text{Cov}[\mathbf{x}_{t+s}, \mathbf{x}_t] \in \mathbb{R}^{n \times n}$ for time-lag $s \in \{0, \dots, S\}$. Time-lagged observed covariances $\Lambda(s) \in \mathbb{R}^{p \times p}$ can be computed from $\Pi_s$ as

$$\Lambda(s) := C\Pi_s C^\top + \delta_{s=0} R. \tag{3}$$

An important special case is the classical linear dynamical system (LDS) with $f(\mathbf{x}_t, \eta_t) = A\mathbf{x}_t + \eta_t$, with $\eta_t \sim \mathcal{N}(0, Q)$ and $\Pi_s = A^s \Pi_0$. As we will see below, our SSID algorithm works directly on these time-lagged covariances, so it is also applicable also to generative models with non-Markovian Gaussian latent dynamics, e.g. Gaussian Process Factor Analysis [2].

**Partial observations and missing data:** We treat multiple partial recordings as a missing-data problem– we use $\mathbf{y}_t$ to model all activity measurements across multiple experiments, and assume that at any time $t$, only some of them will be observed. As a consequence, the data-dimensionality $p$ could now easily be comprised of thousands of neurons, even if only small subsets are observed at any given time. We use index sets $\Omega_t \subseteq \{1, \dots, p\}$, where $i \in \Omega_t$ indicates that variable $i$ is observed at time point $t$. We obtain empirical estimates of time-lagged pairwise covariances for variable each pair $(i, j)$ over all of those time points where the pair of variables is jointly observed with time-lag $s$. We define co-occurrence counts $T_{ij}^s = |\{t | i \in \Omega_{t+s} \wedge j \in \Omega_t\}|$.

In total there could be up to $Sp^2$ many co-occurrence counts– however, for SSOs the number of unique counts is dramatically lower. To capitalize on this, we define co-ocurrence groups $F \subseteq \{1, \dots, p\}$, subsets of variables with identical observation patterns: $\forall i, j \in F \ \forall t \leq T : i \in \Omega_t$ iff $j \in \Omega_t$. All element pairs $(i, j) \in F^2$ share the same co-occurence count $T_{ij}^s$ per time-lag $s$. Co-occurence groups are non-overlapping and together cover the whole range $\{1, \dots, p\}$. There might be pairs $(i, j)$ which are never observed, i.e. for which $T_{ij}^s = 0$ for each $s$. We collect variable pairs co-observed at least twice at time-lag s, $\Omega^s = \{(i, j) | T_{ij}^s > 1\}$. For these pairs we can calculate an unbiased estimate of the s-lagged covariance,

$$\text{Cov}[\mathbf{y}_{t+s}^{(i)}, \mathbf{y}_t^{(j)}] \approx \frac{1}{T_{ij}^s - 1} \sum_t \mathbf{y}_{t+s}^{(i)} \mathbf{y}_t^{(j)} := \tilde{\Lambda}(s)_{(ij)}. \tag{4}$$

## 2.2 Expectation maximization for stitching linear dynamical systems

EM can readily be extended to missing data by removing likelihood-terms corresponding to missing data [29]. In the E-step of our stitching-version of EM (sEM), we use the default Kalman filter and smoother equations with subindexed $C_t = C_{(\Omega_t,:)}$ and $R_t = R_{(\Omega_t,\Omega_t)}$ parameters for each time point $t$. We speed up the E-step by tracking convergence of latent posterior covariances, and stop updating these when they have converged [30]– for long $T$, this can result in considerably faster smoothing. For the M-step, we adapt maximum likelihood estimates of parameters $\theta = \{A, Q, C, R\}$. Dynamics parameters $(A, Q)$ are unaffected by SSOs. The update for $C$ is given by

$$
C_{(i,:)} = \left( \sum \mathbf{y}_t^{(i)} \mathrm{E}[\mathbf{x}_t]^T - \frac{1}{|O_i|} \left( \sum \mathbf{y}_t^{(i)} \right) \left( \sum \mathrm{E}[\mathbf{x}_t]^T \right) \right) \tag{5}
$$
$$
\times \left( \sum \mathrm{E}[\mathbf{x}_t \mathbf{x}_t^T] - \frac{1}{|O_i|} \left( \sum \mathrm{E}[\mathbf{x}_t] \right) \left( \sum \mathrm{E}[\mathbf{x}_t]^T \right) \right)^{-1},
$$

where $O_i = \{t | i \in \Omega_t\}$ is the set of time points for which $y_i$ is observed, and all sums are over $t \in O_i$. For SSOs, we use temporal structure in the observation patterns $\Omega_t$ to avoid unnecessary calculations of the inverse in (5): all elements $i$ of a co-occurence group share the same $O_i$.

## 2.3 Scalable subspace-identification with missing data via moment-matching

**Subspace identification:**  Our algorithm (Stitching-SSID, S3ID) is based on moment-matching approaches for linear systems [31]. We will show that it provides robust initialisation for EM, and that it performs more robustly (in the sense of yielding samples which more closely capture empirically measured correlations, and predict missing ones) on non-Gaussian and nonlinear data. For fully observed linear dynamics, statistically consistent estimators for $\theta = \{C, A, \Pi_0, R\}$ can be obtained from $\{\tilde{\Lambda}(s)\}_s$ [20] by applying an SVD to the $pK \times pL$ block Hankel matrix $H$ with blocks $H_{k,l} = \tilde{\Lambda}(k + l - 1)$. For our situation with large $p$ and massively missing entries in $\tilde{\Lambda}(s)$, we define an explicit loss function which penalizes the squared difference between empirically observed covariances and those predicted by the parametrised model (3),

$$
\mathcal{L}(C, \{\Pi_s\}, R) = \frac{1}{2} \sum_s r_s ||\Lambda(s) - \tilde{\Lambda}(s)||^2_{\Omega^s}, \tag{6}
$$

where $|| \cdot ||_\Omega$ denotes the Froebenius norm applied to all elements in index set $\Omega$. For linear dynamics, we constrain $\Pi_s$ by setting $\Pi_s = A^s \Pi_0$ and optimize over $A$ instead of over $\Pi_s$. We refer to this algorithm as 'linear S3ID', and to the general one as 'nonlinear S3ID'. However, we emphasize that only the latent dynamics are (potentially) nonlinear, dimensionality reduction is linear in both cases.

**Optimization via stochastic gradients:**  For large-scale applications, explicit computation and storage of the observed $\tilde{\Lambda}(s)$ is prohibitive since they can scale as $|\Omega^s| \sim p^2$, which renders computation of the full loss $\mathcal{L}$ impractical. We note, however, that the gradients of $\mathcal{L}$ are linear in $\tilde{\Lambda}(s)^{(i,j)} \propto \sum_t \mathbf{y}_{t+s}^{(i)} \mathbf{y}_t^{(j)}$. This allows us to obtain unbiased stochastic estimates of the gradients by uniformly subsampling time points $t$ and corresponding pairs of data vectors $\mathbf{y}_{t+s}, \mathbf{y}_t$ with time-lag $s$, without explicit calculation of the loss $\mathcal{L}$. The batch-wise gradients are given by

$$
\frac{\partial \mathcal{L}_{t,s}}{\partial C_{(i,:)}} = \left( \Lambda(s)_{(i,:)} - \mathbf{y}_{t+s}^{(i)} \mathbf{y}_t^\top \right) N_s^{i,t} C \Pi_s^\top + \left( [\Lambda(s)^\top]_{(i,:)} - \mathbf{y}_t^{(i)} \mathbf{y}_{t+s}^\top \right) N_s^{i,t+s} C \Pi_s \tag{7}
$$

$$
\frac{\partial \mathcal{L}_{t,s}}{\partial \Pi_s} = \sum_{i \in \Omega_{t+s}} C_{(i,:)}^\top \left( \Lambda(s)_{(i,:)} - \mathbf{y}_{t+s}^{(i)} \mathbf{y}_t^\top \right) N_s^{i,t} C \tag{8}
$$

$$
\frac{\partial \mathcal{L}_{t,s}}{\partial R_{ii}} = \frac{\delta_{s0}}{T_{ii}^0} \left( \Lambda(0)_{(i,i)} - \left( \mathbf{y}_t^{(i)} \right)^2 \right), \tag{9}
$$

where $N_s^{i,t} \in \mathbb{N}^{p \times p}$ is a diagonal matrix with $[N_s^{i,t}]_{jj} = \frac{1}{T_{ij}^s}$ if $j \in \Omega_t$, and 0 otherwise.

Gradients scale linearly in $p$ both in memory and computation and allow us to minimize $\mathcal{L}$ without explicit computation of the empirical time-lagged covariances, or $\mathcal{L}$ itself. To monitor performance and convergence for large systems, we compute the loss over a random subset of covariances. The computation of gradients for $C$ and $R$ can be fully vectorized over all elements $i$ of a co-occurence group, as these share the same matrices $N_s^{i,t}$. We use ADAM [32] for stochastic gradient descent,

which combines momentum over subsequent gradients with individual self-adjusting step sizes for each parameter. By using momentum on the stochastic gradients, we effectively obtain a gradient that aggregates information from empirical time-lagged covariances across multiple gradient steps.

## 2.4  How temporal information helps for stitching

The key challenge in stitching is that the latent space inferred by an LDS is defined only up to choice of coordinate system (i.e. a linear transformation of $C$). Thus, stitching is successful if one can align the $C$s corresponding to different subpopulations into a shared coordinate system for the latent space of all $p$ neurons [23] (Fig. 1). In the noise-free regime and if one ignores temporal information, this can work only if the overlap between two sub-populations is at least as large as the latent dimensionality, as shown by [23]. However, dynamics (i.e. temporal correlations) provide additional constraints for the alignment which can allow stitching even without overlap:

Assume two subpopulations $I_1, I_2$ with parameters $\theta^1, \theta^2$, latent spaces $\mathbf{x}^1, \mathbf{x}^2$ and with overlap set $J = I_1 \cap I_2$ and overlap $o = |J|$. The overlapping neurons $\mathbf{y}_t^{(J)}$ are represented by both the matrix rows $C_{J,:}^1$ and $C_{J,:}^2$, each in their respective latent coordinate systems. To stitch, one needs to identify the base change matrix $M$ aligning latent coordinate systems consistently across the two populations, i.e. such that $M\mathbf{x}^1 = \mathbf{x}^2$ satisfies the constraints $C_{(J,:)}^1 = C_{(J,:)}^2 M^{-1}$. When only considering time-instantaneous covariances, this yields $o$ linear constraints, and thus the necessary condition that $o \geq n$, i.e. the overlap has to be at least as large the latent dimensionality [23].

Including temporal correlations yields additional constraints, as the time-lagged activities also have to be aligned, and these constraints can be combined in the *observability* matrix $J$:

$$\mathcal{O}_J^1 = \begin{pmatrix} C_{(J,:)}^1 \\ C_{(J,:)}^1 A^1 \\ \cdots \\ C_{(J,:)}^1 (A^1)^{n-1} \end{pmatrix} = \begin{pmatrix} C_{(J,:)}^2 \\ C_{(J,:)}^2 A^2 \\ \cdots \\ C_{(J,:)}^2 (A^2)^{n-1} \end{pmatrix} M^{-1} = \mathcal{O}_J^2 M^{-1}.$$

If both observability matrices $\mathcal{O}_J^1$ and $\mathcal{O}_J^2$ have full rank (i.e. rank $n$), then $M$ is uniquely constrained, and this identifies the base change required to align the latent coordinate systems.

To get consistent latent dynamics, the matrices $A^1$ and $A^2$ have to be similar, i.e. $MA^1M^{-1} = A^2$, and correspondingly the time-lagged latent covariance matrices $\Pi_s^1, \Pi_s^2$ satisfy $\Pi_s^1 = M\Pi_s^2 M^\top$. These dynamics might yield additional constraints: For example, if both $A^1$ and $A^2$ have unique (and the same) eigenvalues (and we know that we have identified all latent dimensions), then one could align the latent dimensions of $\mathbf{x}$ which share the same eigenvalues, even in the absence of overlap.

## 2.5  Details of simulated and empirical data

**Linear dynamical system:**  We simulate LDSs to test algorithms S3IDand sEM. For dynamics matrices $A$, we generate eigenvalues with absolute values linearly spanning the interval $[0.9, 0.99]$ and complex angles independently von Mises-distributed with zero mean and concentration $\kappa = 1000$, resulting in smooth latent tractories. To investigate stitching-performance on SSOs, we divded the entire population size of size $p = 1000$ into two subsets $I_1 = [1, \ldots p_1], I_2 = [p_2 \ldots p], p_2 \leq p_1$ with overlap $o = p_1 - p_2$. We simulate for $T_m = 50k$ time points, $m = 1, 2$ for a total of $T = 10^5$ time points. We set the $R_{ii}$ such that 50% of the variance of each variable is private noise. Results are aggregated over 20 data sets for each simulation. For the scaling analysis in section 3.2, we simulate population sizes $p = 10^3, 10^4, 10^5$, at overlap $o = 10\%$, for $T_m = 15k$ and 10 data sets (different random initialisation for LDS parameters and noise) for each population size. We compute subspace projection errors between $C$ and $\hat{C}$ as $e(C, \hat{C}) = ||(I - \hat{C}\hat{C}^\top)C||_F / ||C||_F$.

**Simulated neural networks:**  We simulate a recurrent network of 1250 exponential integrate-and-fire neurons [33] (250 inhibitory and $p = 1000$ excitatory neurons) with clustered connectivity for $T = 60k$ time points. The inhibitory neurons exhibit unspecific connectivity towards the excitatory units. Excitatory neurons are grouped into 10 clusters with high connectivity (30%) within cluster and low connectivity (10%) between clusters, resulting in low-dimensional dynamics with smooth, oscillating modes corresponding to the 10 clusters.

**Larval-zebrafish imaging:**  We applied S3ID to a dataset obtained by light-sheet fluorescence imaging of the whole brain of the larval zebrafish [34]. For this data, every data vector $\mathbf{y}_t$ represents

a $2048 \times 1024 \times 41$ three-dimensional image stack of of fluorescence activity recorded sequentially across 41 z-planes, over in total $T = 1200$ time points of recording at 1.15 Hz scanning speed across all z-planes. We separate foreground from background voxels by thresholding per-voxel fluorescence activity variance and select $p = 7,828,017$ voxels of interest ($\approx 9.55\%$ of total) across all z-planes, and z-scored variances.

## 3 Results

### 3.1 Stitching on simulated data

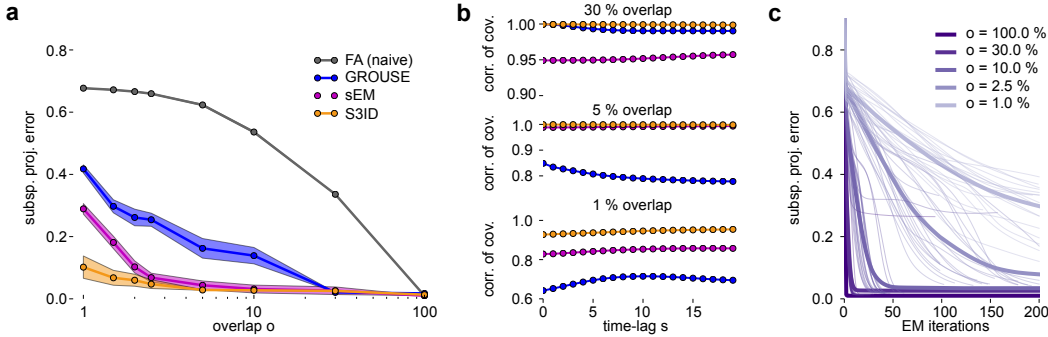

Figure 2: **Dimensionality reduction for multiple partial recordings** **a**) Simulated LDS with $p = 1K$ neurons and $n = 10$ latent variables, two subpopulations, varying degrees of overlap $o$. **a**) Subspace estimation performance for S3ID, sEM and reference algorithms (GROUSE and naive FA). Subspace projection errors averaged over 20 generated data sets, $\pm 1$ SEM. S3ID returns good subspace estimates across a wide range of overlaps. **b**) Estimation of dynamics. Correlations between ground-truth and estimated time-lagged covariances for unobserved pair-wise covariances. **c**) Subspace projection error for sEM as a function of iterations, for different overlaps. Errors per data set, and means (bold lines). Convergence of sEM slows down with decreasing overlap.

To test how well parameters of LDS models can be reconstructed from high-dimensional partial observations, we simulated an LDS and observed it through two overlapping subsets, parametrically varying the size of overlap between them from $o = 1\%$ to $o = 100\%$.

As a simple baseline, we apply a 'naive' Factor Analysis, for which we impute missing data as 0. GROUSE [21], an algorithm designed for randomly missing data, recovers a consistent subspace for overlap $o = 30\%$ and greater, but fails for smaller overlaps. As sEM (maximum number of 200 iterations) is prone to get stuck in local optima, we randomly initialise it with 4 seeds per fit and report results with highest log-likelihood. sEM worked well even for small overlaps, but with increasingly variable results (see Fig. 2c). Finally, we applied our SSID algorithm S3ID which exhibited good performance, even for small overlaps.

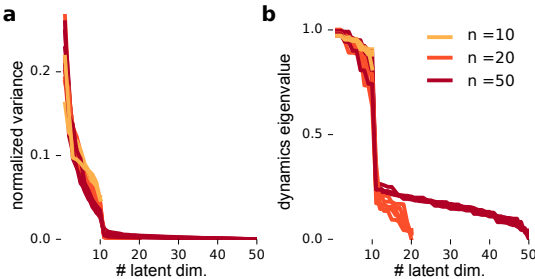

Figure 3: **Choice of latent dimensionality** Eigenvalue spectra of system matrices estimated from simulated LDS data with $o = 5\%$ overlap and different latent dimensionalities $n$. **a**) Eigenvalues of instantaneous covariance matrix $\Pi_0$. **b**) Eigenvalues of linear dynamics matrix $A$. Both spectra indicate an elbow at real data dimensionality $n = 10$ when S3ID is run with $n \geq 10$.

To quantify recovery of dynamics, we compare predictions for pairwise time-lagged covariances between variables not co-observed simultaneously (Fig. 2b). Because GROUSE itself does not capture temporal correlations, we obtain estimated time-lagged correlations by projecting data $\mathbf{y}_t$ onto the obtained subspace and extract linear dynamics from estimated time-lagged latent covariances. S3ID is optimized to capture time-lagged covariances, and therefore outperforms alternative algorithms.

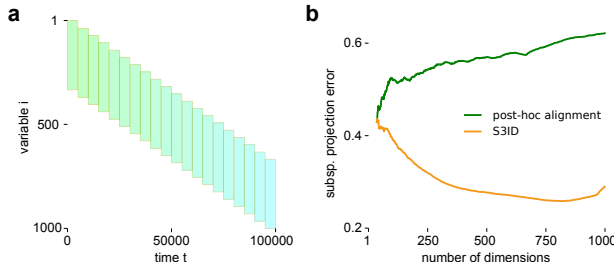

Figure 4: **Comparison with post-hoc alignment of subspaces a**) Multiple partial recordings with 20 sequentially recorded subpopulations. **b**) We apply S3ID to the full population, as well as factor analysis to each of these subpopulations. The latter gives 20 subspace estimates, which we sequentially align using subpopulation overlaps.

When we use a latent dimensionality ($n = 20, 50$) larger than the true one ($n = 10$), we observe 'elbows' in the eigen-spectra of instantaneous covariance estimate $\Pi_0$ and dynamics matrix $A$ located at the true dimensionality (Fig. 3). This observation suggests we can use standard techniques for choosing latent dimensionalities in applications where the real $n$ is unknown. Choosing $n$ too large or too small led to some decrease in prediction quality of unobserved (time-lagged) correlations. Importantly though, performance degraded gracefully when the dimensionality was chosen too big: For instance, at 5% overlap, correlation between predicted and ground-truth unobserved instantaneous covariances was 0.99 for true latent dimensionality $n = 10$ (Fig. 2b). At smaller $n = 5$ and $n = 8$, correlations were 0.69 and 0.89, respectively, and for larger $n = 20$ and $n = 50$, they were 0.97 and 0.96. In practice, we recommend using $n$ larger than the hypothesized latent dimensionality.

S3ID and sEM jointly estimate the subspace $C$ across the entire population. An alternative approach would be to identify the subspaces for the different subpopulations via separate matrices $C_{(I,:)}$ and subsequently align these estimates via their pairwise overlap [23]. This works very well on this example (as for each subset there is sufficient data to estimate each $C_{I,:}$ individually). However, in Fig. 4 we show that this approach performs suboptimally in scenarios in which data is more noisy or comprised of many (here 20) subpopulations. In summary, S3ID can reliably stitch simulated data across a range of overlaps, even for very small overlaps.

## 3.2 Stitching for different population sizes: Combining S3ID with sEM works best

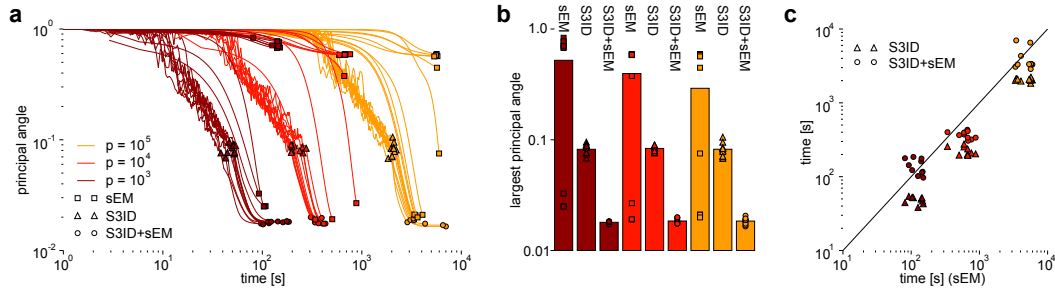

Figure 5: **Initializing EM with SSID for fast and robust convergence** LDS with $p = 10^3, 10^4, 10^5$ neurons and $n = 10$ latent variables, 10% overlap. **a)** Largest principal angles as a function of computation time. We compare randomly initalised sEM with sEM initialised from S3ID after a single pass over the data. **b)** Comparison of final subspace estimate. We can combine the high reliability of S3ID with the low final subspace angle of EM by initialising sEM with S3ID. **c)** Comparison of total run-times. Initialization by S3ID does not change overall runtime.

The above results were obtained for fixed population size $p = 1000$. To investigate how performance and computation time scale with population size, we simulate data from an LDS with fixed overlap $o = 10\%$ for different population sizes. We run S3ID with a single pass, and subsequently use its final parameter estimates to initialize sEM. We set the maximum number of iterations for sEM to 50, corresponding to approximately 1.5h of training time for $p = 10^5$ observed variables. We quantify the subspace estimates by the largest principal angle between ground-truth and estimated subspaces.

We find that the best performance is achieved by the combined algorithm (S3ID + sEM, Fig. 5a,b). In particular, S3ID reliably and quickly leads to a reduction in error (Fig. 5a), but (at least when capped at one pass over the data), further improvements can be achieved by letting sEM do further 'fine-

tuning' of parameters from the initial estimate [35]. When starting sEM from random initializations, we find that it often gets stuck in local minima (potentially, shallow regions of the log-likelihood). While convergence issues for EM have been reported before, we remark that these issues seems to be much more severe for stitching. We hypothesize that the presence of two potential solutions (one for each observation subset) makes parameter inference more difficult.

Computation times for both stitching algorithms scale approximately linear with observed population size $p$ (Fig. 5c). When initializing sEM by S3ID, we found that the cose of S3IDis amortized by faster convergence of sEM. In summary, S3ID performs robustly across different population sizes, but can be further improved when used as an initializer for sEM.

### 3.3 Spiking neural networks

How well can our approach capture and predict correlations in spiking neural networks, from partial observations? To answer this question, we applied S3ID to a network simulation of inhibitory and excitatory neurons (Fig. 6a), divided into into 10 clusters with strong intra-cluster connectivity. We apply S3ID-initialised sEM with $n = 20$ latent dimensions to this data and find good recovery of time-instantaneous covariances (Fig. 6b), but poor recovery of long-range temporal interactions. Since sEM assumes linear latent dynamics, we test whether this is due to a violation of the linearity assumption by applying S3ID with nonlinear latent dynamics, i.e. by learning the latent covariances $\Pi_s$, $s = 0, \ldots, 39$. This comes at the cost of learning 40 rather than $2\, n \times n$ matrices to characterise the latent space, but we note that this here still amounts to only $76.2\%$ of the parameters learned for $C$ and $R$. We find that the nonlinear latent dynamics approach allows for markedly better predictions of time-lagged covariances (Fig. 6b).

We attempt to recover cluster membership for each of the neurons from the estimated emission matrices $C$ using K-means clustering on the rows of $C$. Because the 10 clusters are distributed over both subpopulations, this will only be successful if the latent representations for the two subpoplations are sufficiently aligned. While we find that both approaches can assign most neurons correctly, only the nonlinear version of S3ID allows correct recovery for every neuron. Thus, the flexibility of S3ID allows more accurate reconstruction and prediction of correlations in data which violates the assumptions of linear Gaussian dynamics.

We also applied dynamics-agnostic S3ID when undersampling two out of the ten clusters. Prediction of unobserved covariances for the undersampled clusters was robust down to sampling only 50% of neurons from those clusters. For 50/40/30% sampling, we obtained correlations of instantaneous covariances of 0.97/0.80/0.32 for neurons in the undersampled clusters. Correlation across all clusters remained above 0.97 throughout. K-means on the rows of learned emission matrix $C$ still perfectly identified the ten clusters at 40% sampling, whereas below that it fused the undersampled clusters.

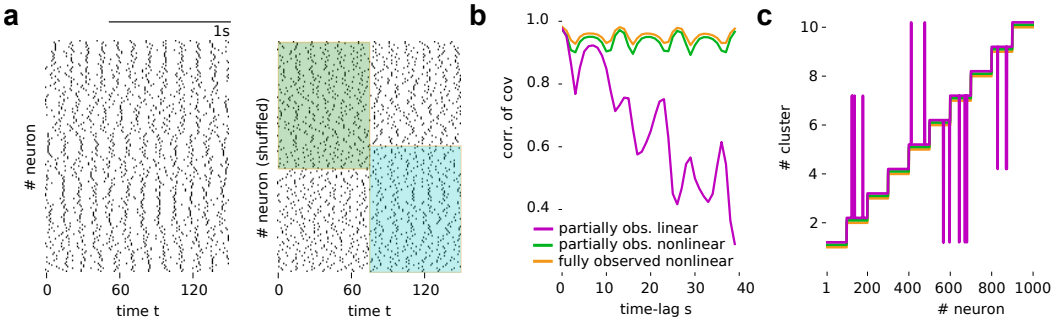

Figure 6: **Spiking network simulation a**) Spiking data for 100 example neurons from 10 clusters, and two observations with $10\%$ overlap (clusters shuffled across observations-subsets). **b**) Correlations between ground-truth and estimated time-lagged covariances for non-observed pairwise covariances, for S3ID with or without linearity assumption, as well as for sEM initialised with linear S3ID. **c**) Recovery of cluster membership, using K-means clustering on estimated $C$.

### 3.4 Zebrafish imaging data

Finally, we want to determine how well the approach works on real population imaging data, and test whether it can scale to millions of dimensions. To this end, we apply (both linear and nonlinear) S3ID

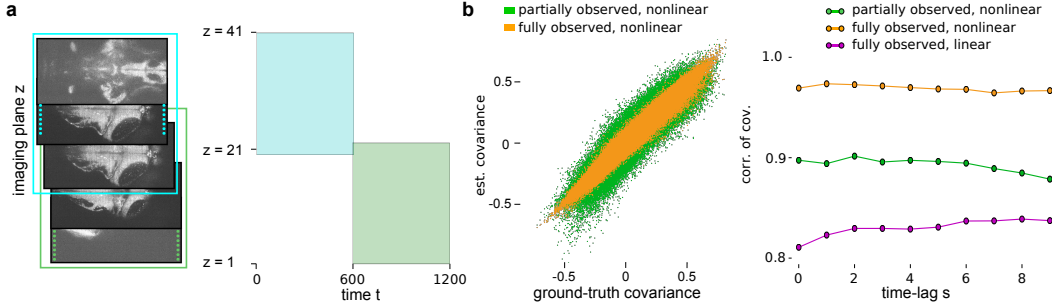

Figure 7: **Zebrafish imaging data** Multiple partial recordings for $p = 7,828,017$-dimensional data from light-sheet fluoresence imaging of larval zebrafish. Data vectors represent volumetric frames from 41 planes. **a**) Simulated observation scheme: we assume the imaging data was recorded over two sessions with a single imaging plane in overlap. We apply S3ID with latent dimensionality $n = 10$ with linear and nonlinear latent dynamics. **b**) Quantification of covariance recovery. Comparison of held-out ground-truth and estimated instantaneous covariances, for $10^6$ randomly selected voxel pairs not co-observed under the observation scheme in **a**. We estimate covariances from two models learned from partially observed data (green: dynamics-agnostic; magenta: linear dynamics) and from a control fit to fully-observed data (orange, dynamics-agnostic). **left**: Instantaneous covariances. **right**: Prediction of time-lagged covariances. Correlation of covariances as a function of time-lag.

to volume scans of larval zebrafish brain activity obtained with light-sheet fluorescence microscopy, comprising $p = 7,828,017$ voxels. We assume an observation scheme in which the first 21 (out of 41) imaging planes are imaged in the first session, and the remaining 21 planes in the second, i.e. with only z-plane 21 (234.572 voxels) in overlap (Fig. 7a,b). We evaluate the performance by predicting (time-lagged) pairwise covariances for voxel pairs not co-observed under the assumed multiple partial recording, using eq. 3. We find that nonlinear S3ID is able to reconstruct correlations with high accuracy (Fig. 7c), and even outperforms linear S3ID applied to full observations. FA applied to each imaging session and aligned post-hoc (as by [23]) obtained a correlation of 0.71 for instantaneous covariances, and applying GROUSE to the observation scheme gave correlation 0.72.

## 4 Discussion

In order to understand how large neural dynamics and computations are distributed across large neural circuits, we need methods for interpreting neural population recordings with many neurons and in sufficiently rich complex tasks [12]. Here, we provide methods for dimensionality reduction which dramatically expand the range of possible analyses. This makes it possible to identify dynamics in data with millions of dimensions, even if many observations are missing in a highly structured manner, e.g. because measurements have been obtained in multiple overlapping recordings. Our approach identifies parameters by matching model-predicted covariances with empirical ones– thus, it yields models which are optimized to be realistic generative models of neural activity. While maximum-likelihood approaches (i.e. EM) are also popular for fitting dynamical system models to data, they are not guaranteed to provide realistic samples when used as generative models, and empirically often yield worse fits to measured correlations, or even diverging firing rates.

Our approach readily permits several possible generalizations: First, using methods similar to [35], it could be generalized to nonlinear observation models, e.g. generalized linear models with Poisson observations. In this case, one could still use gradient descent to minimize the mismatch between model-predicted covariance and empirical covariances. Second, one could impose non-negativity constraints on the entries of $C$ to obtain more interpretable network models [36]. Third, one could generalize the latent dynamics to nonlinear or non-Markovian parametric models, and optimize the parameters of these nonlinear dynamics using stochastic gradient descent. For example, one could optimize the kernel-function of GPFA directly by matching the GP-kernel to the latent covariances.

**Acknowledgements** We thank M. Ahrens for the larval zebrafish data. Our work was supported by the caesar foundation.

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
