[Reviews · NeurIPS 2017]

Reviewer 1



The paper proposes a novel method for extracting a low-dimensional latent subspace to describe high-dimensional measurements when the measurements are of disjoint subsets with limited overlap in time. The author’s propose a variant of subspace identification (SSI) that works by matching empirical covariances to a parametrized model of the covariance. They ultimately propose a combination of their SSI method and EM for application and demonstrate the efficiency and accuracy of their method using both simulation experiments and data-based experiments. The paper appears to be well written and contains appropriate references to prior work. I found the prose and basic technical ideas of the paper surprisingly easy to follow considering the complexity of the material. The work does not propose any big new ideas but it does propose a novel solution to a problem of growing importance in neuroscience. Furthermore, the authors demonstrate the improvement of their method on existing approaches. I can see the method being potentially widely used in neural data analysis. Two points stand out to me as open questions not addressed by the authors. 1) How should one choose the latent dimensionality, and 2) what happens in their simulation experiments when the latent dimensionality is wrong?

Reviewer 2



The authors suggest a method to estimate latent low dimensional dynamics from sequential partial observations of a population. Using estimates of lagged covariances, the method is able to reconstruct unobserved covariances quite well. The extra constraints introduced by looking at several lags allow for very small overlaps between the observed subsets. This is an important work, as many imaging techniques have an inherent tradeoff between their sampling rate and the size of the population. A recent relevant work considered this from the perspective of inferring connectivity in a recurrent neural network (parameters of a specific nonlinear model with observations of the entire subset) [1]. The results seem impressive, with an ability to infer the dynamics even for very low overlaps, and the spiking network and zebrafish data indicate that the method works for certain nonlinear dynamics as well. A few comments: 1. In Figure 3b the error increases eventually, but this is not mentioned. 2. Spiking simulation – the paper assumes a uniform sampling across clusters. How sensitive is it to under-sampling of some clusters? 3. The zebrafish reconstruction of covariance. What is the estimated dimensionality of the data? The overlap is small in terms of percentage (1/41), but large in number of voxels (200K). Is this a case where the overlap is much larger than the dimensionality, and the constraints from lagged covariances do not add much? This example clearly shows the computational scaling of the algorithm, but I’m not sure how challenging it is in terms of the actual data. [1] Soudry, Daniel, et al. "Efficient" shotgun" inference of neural connectivity from highly sub-sampled activity data." PLoS computational biology 11.10 (2015): e1004464.

Reviewer 3



Summary --------- The authors propose a method for stitching together multiple datasets by estimating a low-dimensional latent dynamical system underlying the partially observed neural data. They apply their method to simulated data and a light-sheet imaging dataset from zebrafish. Questions ---------- For predicting unobserved covariances, how well would a matrix completion algorithm work for, say, the zebrafish dataset in Fig 6? It is hard to interpret the correlation of ~0.9 for the partially observed technique without comparison to other baselines. How does the technique compare with the approach of Turaga et. al. on smaller datasets? Minor comments ----------------- - line 35: typo- "Our methods allow *us* to capture the ..." - Font sizes in all figures are too small, making them hard to read - Line markers (triangles and squares) in Fig4a are small and make it hard to compare the different methods - line 263: s/remaining/last - line 271: change "We here provide" to "Here, we provide"